# Nursing Interventions in the Perioperative Pathway of the Patient with Breast Cancer: A Scoping Review

**DOI:** 10.3390/healthcare11121717

**Published:** 2023-06-12

**Authors:** Mafalda Martins Cardoso, Cristina Lavareda Baixinho, Gilberto Tadeu Reis Silva, Óscar Ferreira

**Affiliations:** 1Nursing School of Lisbon, 1600-190 Lisbon, Portugal; crbaixinho@esel.pt (C.L.B.); oferreira@esel.pt (Ó.F.); 2Nursing Research, Innovation and Development Centre of Lisbon (CIDNUR), 1900-160 Lisbon, Portugal; 3Stricto-Sensu Graduate Program at the School of Nursing, Federal University of Bahia, Salvador 40170-110, Brazil; gilberto.tadeu@ufba.br

**Keywords:** cancer patient, breast cancer, nursing interventions, perioperative care, perioperative nursing consultation

## Abstract

The decrease in average hospitalisation time and the increase in outpatient surgery in some types of breast cancer represent gains for the reduction of the negative impact of hospitalisation in women with breast cancer but are also a challenge for the organisation of nursing care to prepare women for surgery, reduce anxiety about the interventions, and ensure continuity of care in the postoperative period. The aim of this study is to identify nursing interventions present in the care provided to patients with breast cancer during the perioperative period. A scoping review was the method chosen to answer the research question: What are the specialised nursing interventions in the perioperative pathway of the patient with breast cancer? Inclusion and exclusion criteria were defined for the articles that were identified in the CINAHL and MEDLINE databases; later, additional sources were identified from the list of bibliographic references for each selected study. The final bibliographical sample consisted of seven articles, which allowed the identification of three key moments of nursing interventions in the perioperative period of patients with breast cancer: the preoperative consultation, the reception of the patient in the operating room, and the postoperative consultation. Factors such as psychological, emotional, and spiritual support, communication and patient-centred care, health education and surgical safety, and the definition of a perioperative pathway for these patients contribute significantly to patients’ satisfaction and the improvement of their quality of life. The results of this study make it possible to establish recommendations for practise and for research, increasing the range of nurses’ actions.

## 1. Introduction

Cancer is the second leading cause of death worldwide, and it is the second leading cause of death in Portugal [1]. The impact of this disease is not only felt by the patient but also represents an emotional and economic burden for the family and for society [2]. Breast cancer, specifically, is the most commonly diagnosed type of cancer and the second leading cause of death in women in Portugal [3]. The existence of population-based screenings with defined regularity that allow the diagnosis of pre-malignant lesions in their early stages leads to an increase in positive prognoses and a decrease in the mortality rate due to breast cancer. The most common therapeutic approach is surgery, sometimes associated with neoadjuvant or adjuvant treatments depending on the type and aggressiveness of the disease [4].

The therapeutic pathway of the patient with breast cancer consists of different stages, with the perioperative period being one of the treatment phases that implies a great effort on the part of the patient and her family and requires a good support network, a supportive health team, and comprehensive access to the health care system and community resources [5].

According to the Portuguese Association of Operating Room Nurses (AESOP), the perioperative period is the entirety of the preoperative, intraoperative, and postoperative moments, and perioperative care is a series of activities developed by perioperative nurses that tend to respond to the needs of the patient undergoing surgery and/or any invasive procedure [6].

Health professionals involved in the treatment of these patients must be aware of their pathway, demonstrating knowledge of the technical and scientific basis for the therapeutic options, thus contributing to a better adjustment of this person to their health/disease process. This knowledge also allows nurses to positively collaborate in the reorganisation of care, of information, and of the pathway itself through their active participation in institutional dynamics [7].

A study that aimed to examine under what conditions a patient might feel adequately prepared to go home and thereby be less likely to rate their length of stay as too short observed that one challenge in allowing patients to feel sufficiently informed and ready to go home is the reduced time for face-to-face consultations [8].

International recommendations advocate that all people proposed for surgery may have prehabilitation, right from the moment of the identified need for surgery, which enables the physical and psychological assessment of the person with the aim of creating a reference profile of the person’s functional state, allowing the identification of deficits that can be solved or attenuated, preventing future complications [9]. The objectives of this process are to increase exercise capacity, increase muscular mass, improve nutritional status, and prepare the psychological condition [10,11].

Studies prove that the implementation of these programmes should start as soon as the person is proposed for surgery; however, for there to be results with the interventions implemented, the programme should last for 4 weeks [9,10,11].

A study that aimed to assess the feasibility and acceptability of an individualised, home-based prehabilitation intervention prior to breast cancer surgery and to explore the potential benefit of prehabilitation on physical fitness and participant-reported physical and psychosocial well-being over time observed that these may facilitate postoperative recovery, impact health behaviour change in the preoperative and postoperative periods, and improve physical activity levels and functional capacity both preoperatively and postoperatively [11].

It is consensual that the adequate preparation of these women has benefits for the reduction of costs associated with treatments while simultaneously keeping an important source of patients’ satisfaction constant [6,7,8].

Based on the arguments mentioned above, the aim of this study is to identify nursing interventions in the care pathway of patients with breast cancer through their perioperative period.

## 2. Materials and Methods

### 2.1. Study Design

According to the aim of this study and due to the exploratory research carried out on the subject, it was considered that the scoping review was the appropriate method for this study as it constitutes “an ideal tool to determine the scope or coverage of a body of literature on a given topic and give a clear indication of the volume of literature and studies available as well as an overview (broad or detailed) of its focus” [12].

The protocol followed the six steps recommended for this type of systematic review: (1) identification of the review question; (2) designation of inclusion and exclusion criteria for studies and identification of relevant studies; (3) selection of studies to be included; (4) assessment of the level of evidence of the collected literature according to the JBI guidelines; (5) discussion of the results; and (6) synthesis and presentation of the obtained results [12,13,14].

Based on the question, “What nursing interventions are described in the literature as relevant in the care pathway of patients with breast cancer, through their perioperative period?” It was possible to define the eligibility criteria and the research strategy, with the perioperative period established as preoperative, intraoperative, and postoperative moments, more specifically during the preoperative consultation, the reception to the operating room, and the postoperative consultation. The concern with these 3 moments is due to the fact that, tendentially, an increasingly larger segment of surgeries is performed on an outpatient regimen or with a clear reduction in hospitalisation time, which implies a (re)organisation of care in order to monitor and intervene alongside these women through all these key moments of nursing contact.

### 2.2. Eligibility Criteria

The use of the acronym PCC (population, concept, and context) allowed the development of the previously presented research question. Each element of the acronym guided the definition of each specific inclusion criteria, presented in Table 1.

### 2.3. Data Collection

The research was carried out using the EBSCOhost database aggregator platform, specifically the CINAHL and MEDLINE databases, in November 2022. The search terms used were breast neoplasms, patients, patient-centred care, nursing interventions, perioperative care, and perioperative period, terms originally retrieved from the Health Sciences Descriptors site and indexed in each of the databases used. The Boolean operators OR and AND were used to operationalize the search, and language filters were applied for full text in Portuguese or English, excluding articles prior to 2017.

Table 2 presents the strategy used in the Medline database.

The search strategy included the use of ‘natural’ and indexed language. In CINHAL’s search strategy, mesh is replaced by subject headings.

The screening of articles by title, abstract, and full article reading was carried out by two independent reviewers; a third element was activated in situations of non-consensus or doubt.

In a second phase, additional sources were identified from the list of bibliographical references for each selected study. A search was also carried out on Google Scholar and open-access scientific repositories using keywords and indexing terms.

### 2.4. Data Processing and Analysis

The researchers built an Excel table, shared in the cloud, to record the content extracted from the articles in the final bibliographic sample: identification of the title of the article/work; author(s), year of publication; type of article; objective(s), method, and main results/conclusions.

The articles that answered the research question and respected the inclusion criteria were subject to analysis, and a narrative synthesis of the results was carried out.

## 3. Results

Figure 1 shows the PRISMA flowchart that describes the selection process, starting with 23 articles that were initially identified by applying the search terms in the databases, all of them from the CINAHL base, and 20 articles from searches performed on platforms such as Google Scholar and the Epistemonikos website. The final sample was comprised of seven articles that met the eligibility criteria and answered the research question.

Below, Table 3 shows the studies that comprise the final bibliographic sample [15,16,17,18,19,20,21].

Two systematic reviews of literature were identified [15,21]: a qualitative study [18], a methodological study [20]; a mixed descriptive study [16]; a correlational predictive study [19]; and a quasi-experimental study [17].

Of the seven selected studies, two refer to the Brazilian population [18,20]; one focuses on the Australian population [16]; another on Iranian women [19]; another on the Turkish population [17]; one in the US [15]; and one in Nigeria [21].

The content analysis of the included articles enabled the narrative synthesis that was organised according to the objectives into 3 points: nursing interventions throughout the preoperative consultation; nursing interventions during the patient’s reception in the operating room; and nursing interventions throughout the postoperative consultation.

### 3.1. Nursing Interventions throughout the Preoperative Consultation

Regarding this perioperative moment, the interventions suggested in the selected articles refer specifically to aspects considered throughout the preoperative nursing consultation, focusing on knowledge about this stage of the breast cancer treatment, offering understanding for the person’s situation, revealing an empathetic attitude, and providing psychological support. It is given great relevance to practical aspects of the preoperative preparation, such as the choice of a supportive bra, options related to breast implants, and the availability of support groups [16]. These aspects are reinforced by one other article [18], which points to the importance of attentive listening as a characteristic of communication with these patients and recommends a humanistic attitude, defined by emotional support and respect for the person’s spirituality. This article also informs readers about the relevance of providing information about the perioperative pathway and the clarification of doubts related to surgical safety, such as regular medication or surgical preparation. It also advises the elaboration of guides or other informative strategies concerning the perioperative pathway, aiming to support patient education on the matter of preparing them for surgery and recovery.

Two articles [15,17] focus on the prevention of complications associated with immobility, recommending the preoperative assessment of the patients’ physical condition, the evaluation of body mass index (BMI), an examination of their posture and range of mobility of the upper limbs, and the documentation of the patient’s routine physical activity. They emphasise the importance of the nurses’ role in education about the benefits of exercise and prehabilitation while preparing for surgery and as agents for motivation and self-efficacy, all important factors promoting recovery in the postoperative period. Both articles aim to prevent complications such as lymphedema and improve the quality of life of these patients.

One article discusses a model of person-centred communication as a fundamental tool for creating a relationship of trust, guaranteeing spiritual and psychological support, promoting the engagement of patients in the decision-making processes, and conducing to better outcomes when confronted with self-image problems [19].

Lastly, one article provides a list of nursing interventions, resorting to the NANDA-I taxonomy (International’s Nursing Diagnosis), allowing us to confer the activities planned for each nursing intervention as well as the results that can be expected [20].

### 3.2. Nursing Interventions during the Patient’s Reception in the Operating Room

Of the totality of the articles selected for this review, only one makes a brief reference to the moment of the patient’s reception in the operating room and the importance of nursing interventions during this transition of care [21]. The article consists of a systematic review of the literature on the effects of non-pharmacological interventions in the reduction of preoperative anxiety, referring to an article [22] that reports on the impact of music therapy in the reduction of anxiety when used during the patient’s transference to the operating room.

### 3.3. Nursing Interventions troughout the Postoperative Consultation

Regarding the postoperative consultation moment, the selected articles inform about the importance of the nurse’s role in meeting the emotional and physical needs of the patients [16]. Two studies [15,17] refer to the follow-up consultation after breast surgery with the purpose of promoting physical exercise and preventing the risks of immobility and the formation of lymphedema, demonstrating positive results in improving the quality of life of patients who comply with the recommended exercise schemes.

The article dedicated to person-centred communication [19] focuses on the impact of this style of communication on the self-image perception of women undergoing mastectomy, revealing benefits linked to patient empowerment and stating the importance of training nurses in this type of communication.

One article reveals that music therapy and acupuncture are significant resources for controlling postoperative pain in patients undergoing breast surgery [21].

## 4. Discussion

This scoping review allowed the mapping of nursing interventions in the perioperative period of patients with breast cancer, identifying the main nursing interventions throughout preoperative and postoperative nursing consultations as well as during the reception to the operating room.

In this regard, the nursing interventions here identified should be added to those already existing in preoperative consultation protocols, namely the importance given to the prevention of postoperative lymphedema in women undergoing axillary dissection, such as the assessment of the physical state, physical activity, and the perimeters of the upper limbs, and the training of exercises recommended for the first phase of the postoperative period [15,17]. These interventions must subsequently be reassessed and reinforced in the postoperative consultation, documented in the patient’s clinical file, and made available to the other members of the health team to ensure continuity of care [20].

The emphasis on physical exercise as a rehabilitation mechanism needs to be prioritised and schooled, not only among patients but also among health professionals, so that it can be included and considered relevant for patients’ education, allowing it to reveal an effective impact on motivation and a change in behaviours [23,24].

In general, the reduction of anxiety seems to be one of the main objectives of nursing interventions in the perioperative period, contributing in a valuable way to patients’ satisfaction with the care provided and to a more positive perception of well-being [18,21,22,23]. A wide variety of approaches aiming to reduce anxiety have recently been recommended for the perioperative context, including education and training in relaxation techniques, deep breathing exercises, meditation, yoga, and music therapy [25].

Psychological and emotional support are also widely mentioned in the selected articles, formulating nursing interventions to be considered in all approaches to patients and, in particular, to patients with breast cancer, taking into account the impact of the surgery on these women’s lives and self-image as a cause of disruption in the person’s self-concept as a woman, mother, or wife in both social and professional contexts [19]. The involvement of the family in the context of care is also one of the needs expressed by women, family members, and caregivers who must be included in the care plan [26].

While attending to the patient, it is essential to ensure the cooperation of a cohesive and communicative multidisciplinary team that shares relevant information to guarantee an effective and satisfactory pathway for the patient, the family, and the health team. From the patients’ point of view, there are several benefits to this interprofessional collaboration in order to promote the success of the treatments, since effective teamwork tends to increase patient satisfaction, improve access to health care, and favour health outcomes [27].

In the perioperative context, it is essential to have a preoperative nursing consultation aimed at women with breast cancer, oriented towards the main differences in the staging and treatment options, with knowledge of the main complications and the skills to identify the desired health outcomes; to be able to welcome the patient to the operating room in a humanised and personal environment, meeting the emotional, psychological, and spiritual needs of this person; and to hold a post-operative consultation sensitive to the pathway travelled, aiming for the satisfaction of the patient and her family, the reduction of their anxiety, and efficiently identifying and promptly resolving problems. Preoperative health education, accomplished through nursing consultations (in person or over the phone), focusing on the patient, adjusting the nurse’s attention to the patient’s individual needs, promoting bidirectional communication, encouraging shared decision-making, and ensuring adequate training of nurses in communication skills and techniques are some of the key factors for optimising perioperative practises [28].

Wu et al. [29] investigated to determine the feasibility of multimodal prehabilitation as part of the breast cancer treatment pathway, and the results of the pilot study showed that patients would overwhelmingly like to participate in this programme (81.3%) as part of their breast cancer treatment and that the programme is feasible and can be delivered.

The results of a recent systematic review reinforce the potential of prehabilitation in the preoperative preparation of women with breast cancer to optimise clinical, physical, and psychological outcomes; however, the authors are aware that researchers and healthcare professionals need to have more knowledge about the effects of unimodal and multimodal interventions implemented in the preoperative period and their effects [30].

The existence of a single study that explores the reception of women in the operating room stands out in this review. Future studies should explore the importance of this activity since studies inform us that a structured reception helps to reduce patients’ anxiety, clarify doubts, prepare the discharge process, guarantee continuity of care, and manage expectations regarding the disease and self-care when returning home [21,22,31,32].

The limitations of this review are due to some methodological options, such as restrictions placed on the language and free access to full text, which allowed some articles to be excluded deductively that may have respected the inclusion criteria. It should also be noted that the studies are heterogeneous, and it was decided not to assess their methodological quality.

## 5. Conclusions

The results obtained in this review contribute to the enrichment of knowledge concerning the needs expressed by women with breast cancer in their surgical pathway and also inform about some of the nursing interventions that must be considered throughout the perioperative period.

The information collected in this scoping review completes the knowledge that perioperative nurses already possess on the prevention of surgical risks, increasing the scope of action of these professionals and connecting them to other hospital structures involved in the treatment of these patients, such as in-patient services or specialty consultations such as breast and oncology consultations. Thereby, the role of the specialist nurse in oncology nursing proves to be of great importance in the training of teams and in the identification of the most appropriate interventions for each case on an individual basis.

From the research point of view, it is suggested that the design and evaluation of nursing interventions be done for each of the three moments described: preoperative consultation, reception in the operating room, and postoperative consultation, given that the research did not allow the identification of structured and rigorously evaluated programmes that allow evidence-based recommendations.

## Figures and Tables

**Figure 1 healthcare-11-01717-f001:**
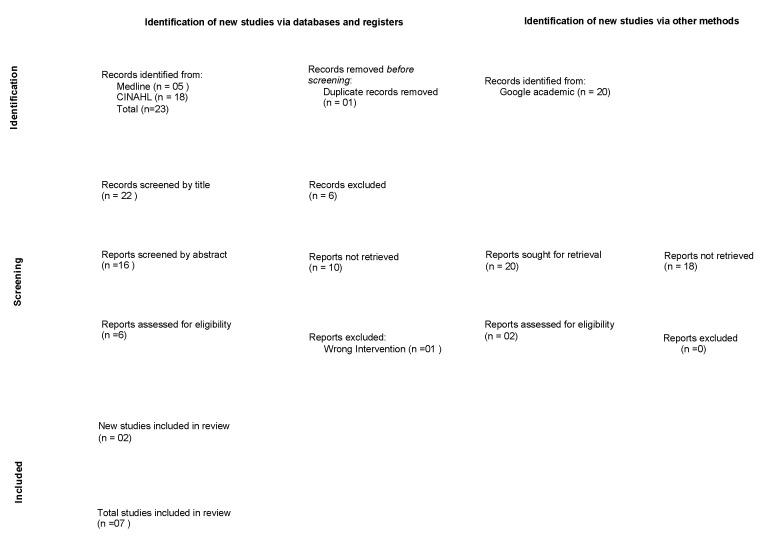
PRISMA-ScR flowchart. Lisbon, 2022.

**Table 1 healthcare-11-01717-t001:** Eligibility criteria according to the PCC acronym. Lisbon, 2022.

PCC	Inclusion Criteria	Exclusion Criteria
P	Adult female (≥19 years) with breast cancer	Teenagers
C	Nursing interventions provided to the patient with breast cancer in the preoperative consultation, in the reception to the operating room, and in the postoperative consultation.	Inpatient nursing interventions. Nursing interventions in late postoperative period
C	Preoperative consultation, reception of the patient in the operating room, and postoperative consultation	Primary health care. Nursing homes. Rehabilitation units.

**Table 2 healthcare-11-01717-t002:** Medline search strategy. Lisbon, 2022.

	Search Strategy	Number Articles
#1	((((((((((((((((adult[Title/Abstract])) OR (elderly[Title/Abstract]))) OR (older person[Title/Abstract])) OR (older people[Title/Abstract])) OR (age[Title/Abstract])) OR (aged[Title/Abstract])) OR (elder*[Title/Abstract])) OR (adult[MeSH Terms])) OR (aged[MeSH Terms])) OR (frail older adult[MeSH Terms]))) AND (woman[Title/Abstract])) NOT (adolescent[Title/Abstract]) Filters: Free full text, from 2017–2022	21,448
#2	((breast cancer[Title/Abstract])) OR (breast cancer[MeSH Terms]) Filters: Free full text, from 2017–2022	76,142
#3	((((((((((((((((nursing[Title/Abstract])) OR (nurs*[Title/Abstract])) OR (interv*[Title/Abstract])) OR (advanced Nurs*[Title/Abstract])) OR (educacional interventions[Title/Abstract])) OR (educat*[Title/Abstract])) OR (capacit*[Title/Abstract])) OR (nursing support[Title/Abstract])) OR (surgical nursing[Title/Abstract])) OR (advanced Nurs*[Title/Abstract])) OR (humanization[Title/Abstract])) OR (advanced practice nursing[MeSH Terms])) OR (early intervention[MeSH Terms])) OR (activities, educational[MeSH Terms])) OR (building, capacity[MeSH Terms])) Filters: Free full text, from 2017–2022	875,933
#4	(((((((((((((consultation[Title/Abstract]) OR (consult*[Title/Abstract])) OR (preoperative consult*[Title/Abstract])) OR (postoperative consult*[Title/Abstract])) OR (admission to the operating room[Title/Abstract])) OR (operating room[Title/Abstract])) OR (ambulatory surgery[Title/Abstract])) OR (operating room[Title/Abstract])) OR (operating theater[Title/Abstract])) OR (care, postoperative[MeSH Terms])) OR (care, preoperative[MeSH Terms])) OR (period, preoperative[MeSH Terms])) OR (ambulatory surgery[MeSH Terms])) OR (ambulatory care facilities, hospital[MeSH Terms])Filters: Free full text, from 2017–2022	44,718
#5	#1 AND #2 AND #3 AND #4	5

**Table 3 healthcare-11-01717-t003:** Final bibliographic sample. Lisbon, 2022.

Study/Country/Year	Study Design and Aim	Results
Wilson [15]. USA (2017)	Systematic review of literature To describe how mobilization stretches and exercise decrease shoulder impairments, a complication related to breast cancer surgery, thus improving quality of life.	Prevent lymphedema. Observe patient’s posture. Measure the circumference of both arms. Assess shoulder range. Measure BMI. Check exercise habits: type of exercise, duration, frequency, and intensity. Educate for exercise in the postoperative period. Reinforce the information about the exercises. Clarify doubts
Brown, Refeld, and Cooper [16]. Australia (2018)	Mixed descriptive study To understand what, if any, differences exist in the perception of a breast care nurse (BCN) consultation between women who experienced a preoperative, face-to-face counselling and education opportunity with a BCN and those who required a telephone consultation or were unable to experience a preoperative BCN consultation.	Offer knowledge and understanding; Psychological support, empathy; Help with practical questions, information about breast implants, supportive bras, and support groups.
Nemli & Kartin [17]. Turkey (2019)	Quasi-experimental study To determine the effects of exercise training that was supported with follow-up calls at home on the postoperative level of physical activity and quality of life of women with breast cancer.	Measurement of the upper limbs; Training of the exercises available in the brochure, clarifying doubts; Reinforce the importance of exercise and clarify doubts;
Trescher, et al. [18]. Brazil (2019)	Qualitative study To know the care needs in the preoperative period for tumour resection in the perception of women with breast cancer and nurses	Attentive listening; emotional support. Transmission of information about the patient’s pathway, clarification of doubts about therapy, and surgical preparation. Design guides and strategies for providing patient orientation. Understanding the spiritual dimension and its place in the treatment of the person.
Ghaffari, et al. [19]. Iran (2020)	Correlational predictive study To determine the predictive values of patient-centred communication (PCC) and patient characteristics on body image (BI) perception in postmastectomy patients.	Build a trusting relationship; Enhance the patient’s involvement in decision-making; Information exchange; Respond to emotional needs; Help manage uncertainty; Support the patient’s independence by providing appropriate resources.
Trescher, et al. Brazil [20]. (2020)	Methodological study Development of a model for nursing consultation in the preoperative period of women with breast cancer at an oncological outpatient.	List of nursing interventions according to the NANDA-I taxonomy; Planned activities; Expected results
Tola, Chow, and Liang [21]. Nigeira (2021)	Systematic review of literature To identify, analyse, and synthesise the effects of non-pharmacological interventions on preoperative anxiety and acute postoperative pain in women undergoing breast cancer surgery.	Use of music therapy to reduce pre- and postoperative anxiety; Postoperative aromatherapy in pain management; Acupuncture to reduce anxiety and pain.

## Data Availability

Data are available only upon request to the authors.

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
