# Peer review of "Nursing Interventions in the Perioperative Pathway of the Patient with Breast Cancer: A Scoping Review"

_healthcare, 2023, doi:10.3390/healthcare11121717_

Round 1
Reviewer 1 Report
Dear Authors,
Thank you for the opportunity to read your manuscript. It raises an important issue not only in Portugal but all over the world. Proper preparation of the patient for perioperative care is one of the elements affecting the effect of surgical treatment.
It is worth writing about prehabitation and its impact on the outcome of treatment, including the patient's well-being.
Table 2 adds nothing, I think it should be removed.
Table 3, I believe, is too poor in information regarding the described manuscript.
Does the first item in table 3 meet the inclusion criteria?
I believe the manuscript is an example of a tautology. It does not bring anything new, it does not systematize knowledge. An important topic, but the study was conducted chaotically.
Minor English language edition required. Easy to understand language, few mistakes.
Author Response
First of all, we would like to thank you for the time spent in reviewing our article. Your suggestions allowed us to introduce some changes that we believe improved the overall quality of the article.
Review 1
Thank you for the opportunity to read your manuscript. It raises an important issue not only in Portugal but all over the world. Proper preparation of the patient for perioperative care is one of the elements affecting the effect of surgical treatment.
It is worth writing about prehabitation and its impact on the outcome of treatment, including the patient's well-being.
We introduced the concept of prehabilitation in the introduction and in the discussion, presenting references of its application in people with oncological disease, namely breast cancer.
Table 2 adds nothing, I think it should be removed.
We decided to maintain table 2 since the PRISMA-ScR guideline recommends the presentation of the research strategy in at least one database.
Table 3, I believe, is too poor in information regarding the described manuscript.
We agree that the table is synthetic. Anyway, we opted to present the minimum information about the articles and, following the table, we organized the nursing interventions mentioned within categories, thus avoiding the duplication of the contents.
Does the first item in table 3 meet the inclusion criteria?
Yes, it does indeed. This relates with the optimal nursing interventions provided to the patient with breast cancer throughout their pathway, starting in the preoperative consultation, and reinforced during the patients stay in the operating room and after, in the postoperative consultation. The analysis of the patients exercise habits and their helps evaluate their potential for recovery and to plan the best strategies for them to comply, preventing complications after the surgery.
I believe the manuscript is an example of a tautology. It does not bring anything new, it does not systematize knowledge. An important topic, but the study was conducted chaotically.
The study followed the JBI protocol and we believe it was conducted rigorously, but the use of only 2 databases limited the results. When you say it doesn’t bring new knowledge, we, on the other hand, see how it helps the discussion of the need to reorganize nursing care for these patients, to ensure that the preoperative preparation is carried out in a different way, a more personal and effective way. It could be an article with potential for developing work with master's students in the oncology nursing field.
Reviewer 2 Report
Dear authors,
Thank you for submitting your work. Please address the following comments raised, make necessary changes and resubmit.
Having a search of only 2 databases is a limitation. A scoping review is also expected to search the grey literature- Google Scholar, Scopus, Cochrane database search could have provided more hits.
Lines 131: A systematic review can not be included for a scoping review.
Prehabilitation is a common practice before oncological surgeries. The authors have not taken into account anything related to this.
1. Brahmbhatt P, et al. Feasibility of Prehabilitation Prior to Breast Cancer Surgery: A Mixed-Methods Study. Front Oncol. 2020 Sep 25;10:571091.
2. Wu F, et al. The Feasibility of Prehabilitation as Part of the Breast Cancer Treatment Pathway. PM R. 2021 Nov;13(11):1237-1246.
3. Toohey K, et al. A systematic review of multimodal prehabilitation in breast cancer. Breast Cancer Res Treat. 2023 Jan;197(1):1-37.
Nursing has a great role in prehabilitation.
ERAS (Enhanced REcovery After Surgery) has been extensively practiced for various oncological surgeries, including breast surgeries, especially in the postoperative period. This deserves a mention in this scoping review.
Offodile AC 2nd, et al. Enhanced recovery after surgery (ERAS) pathways in breast reconstruction: systematic review and meta-analysis of the literature. Breast Cancer Res Treat. 2019 Jan;173(1):65-77.
The language needs minor technical edits, preferably after the revision.
Author Response
First of all, we would like to thank you for the time spent in reviewing our article. Your suggestions allowed us to introduce some changes that we believe improved the overall quality of the article.
Lines 131: A systematic review cannot be included for a scoping review.
The identification of systematic reviews contraindicates the performance of a scoping because it allows realizing that there are already studies on the subject and there is no need to map the literature. However, the included systematic review does not directly answer the research question, only partially, hence the decision to include it.
Prehabilitation is a common practice before oncological surgeries. The authors have not taken into account anything related to this.
We introduced the concept of prehabilitation in the introduction and in the discussion, presenting references of its application in people with oncological disease, namely breast cancer.
We appreciate the article suggestions, which allowed us to improve the introduction and the discussion. We added them to the final list of references.
Round 2
Reviewer 1 Report
Dear Authors,
thank you for clarifying my doubts.
However, I still consider table 2 redundant.
Author Response
Dear review:
The maintenance of table 2 is due to the recommendation of Prisma ScR to present the full electronic search strategy for at least 1 8 database, including any limits used, such that it could be repeated.
We attach the report guideline for confirmation.
We appreciate the time spent reviewing our article and contributions to its improvement.

Reviewer 2 Report
Dear authors,
Thank you for responding to the comments and making changes as necessary at appropriate places.
The English language needs technical edits. Overall, the message is pretty clear.
Author Response
Dear review:
We revised the text and corrected some mistakes identified.
We appreciate the time spent reviewing our article and contributions to its improvement.
Best regards;